# Patient flow within UK emergency departments: a systematic review of the use of computer simulation modelling methods

Syed Mohiuddin,[1,2] John Busby,[3] Jelena Savović,[1,2] Alison Richards,[1,2] Kate Northstone,[1,2] William Hollingworth,[1,2] Jenny L Donovan,[1,2] Christos Vasilakis[4]

► Prepublication history and additional material is available. To view please visit the journal (http://dx.doi.org/ 10.1136/ bmjopen-2016-015007)

[1]NIHR CLAHRC West, University Hospitals Bristol NHS Foundation Trust, Bristol, UK
[2]School of Social and Community Medicine, University of Bristol, Bristol, UK
[3]School of Medicine, Queen's University Belfast, Belfast, UK
[4]Centre for Healthcare Innovation & Improvement (CHI2), School of Management, University of Bath, Bath, UK

**Correspondence to**
Dr Syed Mohiuddin;
syed.mohiuddin@bristol.ac.uk

## ABSTRACT

**Objectives** Overcrowding in the emergency department (ED) is common in the UK as in other countries worldwide. Computer simulation is one approach used for understanding the causes of ED overcrowding and assessing the likely impact of changes to the delivery of emergency care. However, little is known about the usefulness of computer simulation for analysis of ED patient flow. We undertook a systematic review to investigate the different computer simulation methods and their contribution for analysis of patient flow within EDs in the UK.

**Methods** We searched eight bibliographic databases (MEDLINE, EMBASE, COCHRANE, WEB OF SCIENCE, CINAHL, INSPEC, MATHSCINET and ACM DIGITAL LIBRARY) from date of inception until 31 March 2016. Studies were included if they used a computer simulation method to capture patient progression within the ED of an established UK National Health Service hospital. Studies were summarised in terms of simulation method, key assumptions, input and output data, conclusions drawn and implementation of results.

**Results** Twenty-one studies met the inclusion criteria. Of these, 19 used discrete event simulation and 2 used system dynamics models. The purpose of many of these studies (n=16; 76%) centred on service redesign. Seven studies (33%) provided no details about the ED being investigated. Most studies (n=18; 86%) used specific hospital models of ED patient flow. Overall, the reporting of underlying modelling assumptions was poor. Nineteen studies (90%) considered patient waiting or throughput times as the key outcome measure. Twelve studies (57%) reported some involvement of stakeholders in the simulation study. However, only three studies (14%) reported on the implementation of changes supported by the simulation.

**Conclusions** We found that computer simulation can provide a means to pretest changes to ED care delivery before implementation in a safe and efficient manner. However, the evidence base is small and poorly developed. There are some methodological, data, stakeholder, implementation and reporting issues, which must be addressed by future studies.

## INTRODUCTION

An emergency department (ED), also known as accident & emergency department (A&E), provides acute care for patients who attend hospital without prior appointment. The EDs of most hospitals customarily operate 24 hours a day, 7 days a week. Nevertheless, overcrowding in EDs is an increasing problem in countries around the world, and especially so in the UK.[1 2] ED overcrowding has been shown to have many adverse consequences such as increased medical errors,[3] decreased quality of care and subsequently poor patient outcomes,[4] increased workload,[1] frustration among ED staff,[4 5] ambulance diversions,[6] increased patient dissatisfaction,[5] prolonged patient waiting times[7] and increased cost of care.[8] Furthermore, some less severely ill patients may leave without being seen by a physician, only to return later with a more complicated condition.[8]

In the UK, there is an enormous pressure from public and government to alleviate overcrowding and long waiting times experienced in ED.[1] The Department of Health set a target standard for acute hospitals in

BMJ

the National Health Service (NHS) that at least 95% of patients attending an A&E department must be seen, treated, admitted or discharged under 4 hours.[7] This 4-hour target standard was initially set at 98% in 2004, but later reduced to 95% in 2010. Beyond target setting, it has been argued that ED overcrowding can be improved by lean healthcare thinking with a focus on improving patient flow.[1 9]

Over recent decades, computer simulation and other modelling methods have been used to analyse ED patient flow and resource capacity planning.[10–14] In essence, a computer simulation model is a simplified representation of reality used to aid the understanding of the key relationships and dynamics in the care process, and to evaluate the likely impact of changes before implementation. Typically, a simulation model is based on the notion that each simulated individual is tracked through the care process; the population effect is then estimated from the sum of the individual effects.[15]

The precise way in which a simulation model works depends on the type of simulation method used. Generally, simulation models can be categorised as static or dynamic, as stochastic or deterministic, and as discrete time or continuous time.[16] A static simulation represents a process at a particular point in time, whereas a dynamic simulation represents a process as it evolves over time. A simulation model in which at least one input parameter is a random variable is said to be stochastic, whereas a simulation model having no random variables is said to be deterministic. A discrete time model is one in which the state variables change instantaneously at discrete points in time. In contrast, a continuous time model is one in which the state variables change continuously with respect to time. The advancement of computer technology has undoubtedly supported the use of more sophisticated simulation methods for modelling healthcare processes. Today, for example, computer simulation is also capable of providing an insight into the workings of a system through visual animation.

Various types of computer simulation exist, including discrete event simulation (DES), system dynamics (SD) and agent-based simulation (ABS). DES is a widely used method, and can replicate the behaviour of complex healthcare systems over time. A DES model is a network of queues and activities (such as having a blood test, X-ray and treatment). One of the major advantages of using a DES model is its flexibility to model complex scenarios at the individual level. Within a DES model, individuals move from one activity to another in sequential order at a particular point in time. Typically, the individuals enter a system and visit some of the activities (not necessarily only once) before leaving the system. The variables that govern the movement of modelled individuals (such as arrival rate and duration of treatment) can be random and thus readily capture the variation that is inherent in healthcare. As such, a DES model is considered particularly suitable for modelling queuing systems. This simulation method is able to incorporate life histories and complex scenarios at the individual level to influence the care pathway taken and the time between events, but specialist analytical knowledge is required typically to achieve a greater flexibility.[17]

Another widely used simulation method is SD, which is used to understand the behaviour of complex healthcare systems over time through capturing aggregate (instead of individual) flows of patients. An SD model is essentially a collection of stocks and flows between them. Stocks are basic stores of quantities over time, for example, number of patients with a disease or in a particular part of a hospital department. Flows define the movement of objects between different stocks over time. Unlike DES, SD does not lend itself readily to including random variables and thus input parameters are given as simple rates in the majority of studies. As such, SD is considered not the ideal method of choice for modelling a closely focused system that involves resource-constrained queuing networks, such as an ED.[10] In a situation like this, DES should rather be the method of choice to model high level of distinct detail.[18] ABS is another method that has more recently been used in modelling the healthcare systems. As a new method in this application area, ABS is often overlooked in favour of using more established methods of DES and SD. The usefulness and practicalities of ABS in modelling patient flow are not well understood.[19]

Increasing interest in this area is reflected in the number of computer simulation studies of ED patient flow and resource capacity planning that have been published over recent decades. However, little is known about the usefulness of different computer simulation methods for analysis of any changes to the delivery of emergency care. We, therefore, systematically investigated the peer-reviewed literature on the use of computer simulation modelling of patient flow within EDs in the UK. Our specific objectives were as follows: (1) to investigate the contribution that computer simulation studies make to our understanding of the problem of ED overcrowding; (2) to identify the methodology used to conduct patient flow simulation in terms of key assumptions, systems requirements, and input and output data; (3) to assess the usefulness of each simulation method for service redesign and evaluating the likely impact of changes related to the delivery of emergency care; (4) to report on differences in conclusions about ED performance with different simulation modelling methods; and (5) to identify studies that explicitly aimed to meet the prespecified needs of stakeholders.

## METHODS

We conducted a systematic review of the peer-reviewed literature to identify computer simulation studies of patient flow within hospital EDs in the UK. This review complies with the online supplementary PRISMA checklist (www.prisma-statement.org). We produced a review protocol (available from the corresponding author on

request) and set out the process to address our specific objectives.

## Search strategy

We retrieved relevant studies from the following bibliographic databases: MEDLINE, EMBASE, COCHRANE, WEB OF SCIENCE, CINAHL, INSPEC, MATHSCINET and ACM DIGITAL LIBRARY. We used a key review paper[20] to select these databases, which were searched from the date of their inception until 31 March 2016. A search strategy was designed to explore three main domains of knowledge associated with the area of our interest: computer simulation, emergency care and patient flow. We included a wide range of search phrases, both keywords and medical subject headings, such as 'computer simulation,' 'emergency department,' 'patient care,' 'patient flow,' 'waiting time,' 'time to treatment' and 'length of stay.'

We first developed the search strategy for MEDLINE since it is known to allow a rich taxonomy of subjects and rubrics. We used a key review paper[20] to inform the MEDLINE search strategy and made further refinements using other relevant studies to improve sensitivity. Online supplementary appendix 1 shows the MEDLINE search strategy and results from 1946 to end of March 2016. We adapted the MEDLINE strategy to search the other databases (available from the corresponding author on request). We also conducted backward and forward citation searches of all included studies using Google Scholar.

## Inclusion criteria

We identified studies as being eligible for inclusion if they: (1) were published in peer-reviewed journals or conference proceedings as full papers; (2) were conducted within the ED of an established UK NHS hospital responsible for assessing and treating civilians in need of emergency care; (3) captured the progress of patients through at least two activities of an ED care process; and (4) used a computer simulation method such as DES, SD, ABS, hybrid simulation, Monte Carlo simulation, distributed simulation or stochastic modelling.

We excluded editorials, letters, commentaries, conference abstracts, notes and books. We also excluded studies that used methods such as regression analysis, likelihood ratio test, time series analysis, generalised linear model, mathematical programming, optimisation methods, queuing theory, structural equation modelling, process mapping, problem structuring method or risk analysis without combining it with a computer simulation method.

## Selection of studies for full-text review

To identify the studies suitable for full-text review, two authors (SM and JB) independently screened the titles and abstracts of all the initially retrieved studies. The individual responses from each reviewer were stored in a common database. At this stage, a study was excluded if it was clearly irrelevant based on our inclusion criteria. In cases of discrepancy, we selected the studies for full-text review by consensus.

## Appraisal of studies for inclusion

An electronic questionnaire was designed to appraise the studies selected for full-text review. The questionnaire included four key questions: (1) Is the study a full paper published in a peer-reviewed journal or conference proceedings? (2) Is the study set within the UK NHS? (3) Is the study conducted within the ED of an established hospital? (4) Does the study use a computer simulation model of emergency patient flow? A study with positive responses on these four questions was then included in the final review. SM and JB completed this process independently and resolved any discrepancies that arose by consensus. Further to the electronic search, SM and JB reviewed the backward and forward citations of all studies included in the electronic search to identify other potentially relevant studies.

## Data extraction

An electronic data extraction form was created to retrieve information about a number of key aspects, including simulation methods, data sources, key assumptions, input and output data, conclusions drawn and benefits of simulation outputs in practice. SM and JB independently recorded, collated and extracted the necessary information. Any discrepancies were resolved by consensus.

## RESULTS

We retrieved a total of 2436 references from the 8 databases: 437 from MEDLINE; 460 from EMBASE; 14 from COCHRANE; 253 from WEB OF SCIENCE; 65 from CINAHL; 1103 from INSPEC; 4 from MATHSCINET; and 100 from ACM DIGITAL LIBRARY. We removed 440 duplicate references, and then assessed the remaining 1996 unique references by title and abstract screening. At this stage, we selected 159 of the 1996 studies for full-text review. Nineteen of the 159 studies were included following full-text review. Two more studies were included from the backward and forward citation searching of the 19 studies. A total of 21 studies[10–14 21–36] were included in the final review. Four studies[37–40] were excluded from the final review because the models used by these studies are identical to the models reported in other already included studies.[13 30 33 36] Figure 1 shows a summary of the study selection process.

Table 1 summarises the included studies, outlining publication type, simulation purpose, ED details and patient flow description. The first study[33] was published in 2000 and the most recent study[21] was published in 2013. The maximum number of studies (n=4) published in any single year was in 2006 and 2011. Nine of the 21 included studies (43%) were published in conference proceedings. The highest number of studies (n=7) was published in proceedings of the Winter Simulation Conference, the second highest (n=5) was in the Emergency Medicine Journal, and the third highest (n=3)

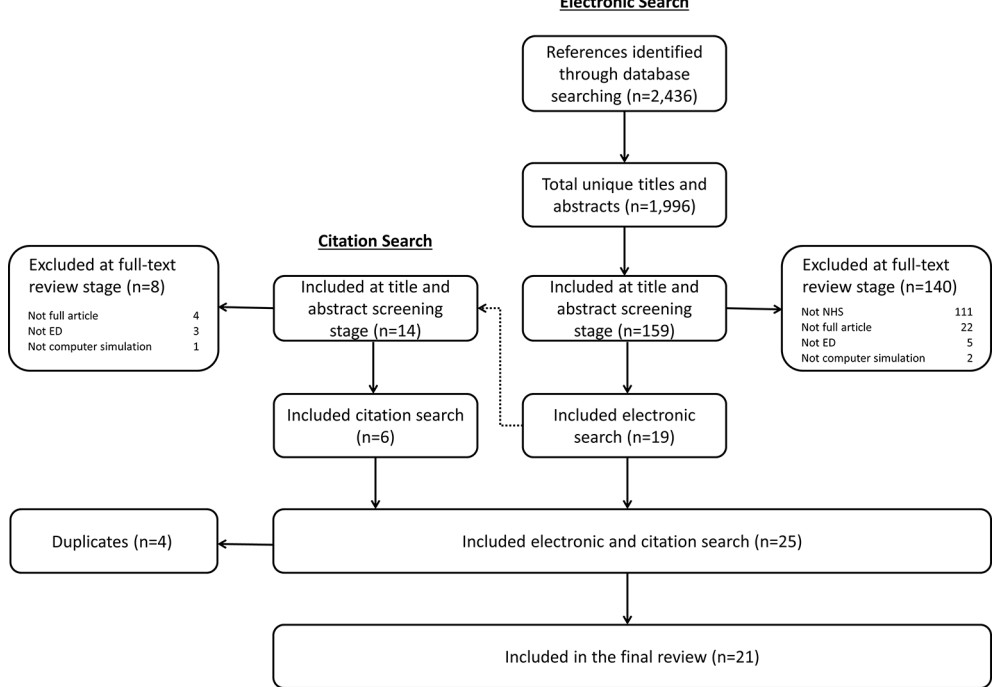

**Figure 1** Flow chart of the study identification and inclusion process. ED, emergency department; NHS, National Health Service.

was in the Journal of the Operational Research Society. More than two-thirds of the studies (n=16; 76%) did not provide the name of the hospital studied. All 21 studies described the underlying purpose of simulation; and in many cases (n=16; 76%), this centred on service redesign. Surprisingly, seven studies[12 13 21 25 31 34 36] (33%) did not provide any details about the ED being investigated, while five studies[10 21 23 26 30] (24%) did not provide patient flow diagram.

Table 2 provides summary of simulation methods, including simulation type, key assumptions and use of software. The types of simulation varied only between two methods (DES and SD). DES modelling was used in 19 studies (90%), while SD was used in 2 studies (10%). All but one study[25] either explicitly or implicitly justified the choice of underlying modelling method used. The majority of studies (n=18; 86%) used specific hospital models of ED patient flow. The reporting of modelling assumptions was poor overall. For example, as many as 12 studies[10 12 13 21 23–25 29 31 32 34 35] (57%) did not provide any details about simulation duration, warm-up period and run number. Only five studies[11 22 27 30 33] (24%) specified the number of simulation runs and three studies[27 30 36] (14%) specified the simulation warm-up period. Simulation duration ranged from 24 hours[27 33] to 52 weeks.[26 30] Almost 50% (n=10) of the studies used Simul8 software (www.simul8.com) for running the model. Two studies[24 34] (10%) did not provide any details about the use of software.

Table 3 provides detail of simulation inputs and outputs. The identified models were populated from three sources of data: primary (ie, collected within the hospital being studied), secondary (ie, collected in another setting) and expert opinion. One study[21] did not describe the source of data for any of the model inputs. Three studies[13 26 31] described the source of patient arrival rates, but not the sources of activity duration, activity progression and use of resources. Eight studies (38%) stated explicitly that they used expert opinion to populate some of the model inputs. The proportion of studies that used primary data was reasonably high (table 3). In particular, 95% (n=20) of the studies used primary data for patient arrival rates, 67% (n=14) for activity duration, 62% (n=13) for activity progression and 52% (n=11) for resource inputs.

The most common changes considered in the simulation studies were ED patient flow (eg, changes in the triage system for arriving patients[30]) and resource capacity planning (eg, changes in the number of cubicles[30]). However, one-third of the studies[11 13 21 24 26 27 31] (n=7) did not provide any details about the changes considered by the simulation. The majority of studies (n=19; 81%) considered patient waiting times (ie, time from arrival to discharge, admission or transfer) as the key outcome measure. In particular, 11 studies[10 11 13 21 22 25 27 29–31 34] considered patient waiting times alone, 7 studies[12 23 24 28 32 35 36] considered patient waiting times and resources used, and the other study[33] considered patient waiting times, resources used and elective cancellations. Two other outcome measures considered were resources used[26] and bed occupancy.[14]

Only 12 studies (57%) reported some involvement of stakeholders in the simulation study, mainly when deciding the study questions or specifying the model structure. However, in the study conducted by Mould

**Table 1** Detail of the included studies

| Study name (year) | Publication type (name) | Hospital name | Simulation purpose | ED Detail | Patient Flow Description |
|---|---|---|---|---|---|
| Anagnostou et al[21] (2013) | Conference proceedings (Winter Simulation Conference) | Unknown several hospitals in Greater London | Proof of concept | No detail | Textual; activity list |
| Au-Yeung et al[22] (2006) | Conference proceedings (Modelling and Simulation) | Unknown hospital in North London | Service redesign | More detail | Flow chart; textual |
| Baboolal et al[23] (2012) | Journal article (Emergency Medicine Journal) | University Hospital of Wales | Service redesign | More detail | Textual |
| Bowers et al[24] (2009) | Journal article (Journal of Simulation) | Unknown hospital in Fife, Scotland | Service redesign | Some detail | Flow chart |
| Brailsford et al[10] (2004) | Journal article (Journal of Operational Research Society) | Nottingham City Hospital and QMC in Nottingham | Service redesign | More detail | Textual |
| Coats and Michalis[25] (2001) | Journal article (Emergency Medicine Journal) | Royal London Hospital in Whitechapel, London | Service redesign | No detail | Flow chart |
| Codrington-Virtue et al[26] (2006) | Conference proceedings (Computer-Based Medical Systems) | Unknown hospital | Understand capacity | More detail | Textual |
| Codrington-Virtue et al[27] (2011) | Conference proceedings (Winter Simulation Conference) | Unknown hospital | Proof of concept | Some detail | Flow chart; textual |
| Coughlan et al[28] (2011) | Journal article (Emergency Medicine Journal) | Unknown district general hospital in West London | Service redesign | Some detail | Flow chart; textual |
| Davies[29] (2007) | Conference proceedings (Winter Simulation Conference) | Unknown hospital | Service redesign | More detail | Flow chart; textual |
| Eatock et al[11] (2011) | Journal article (Journal of Health Org. and Management) | Hillingdon Hospital in West London | Service redesign | More detail | Flow chart; textual |
| Fletcher et al[12] (2007) | Journal article (Journal of Operational Research Society) | Unknown hospitals (n=10) | Service redesign | No detail | Flow chart; textual |
| Günal and Pidd[13] (2009) | Journal article (Emergency Medicine Journal) | Unknown hospital | Understand behaviour | No detail | Flow chart; textual |
| Günal and Pidd[30] (2006) | Conference proceedings (Winter Simulation Conference) | Unknown hospital | Service redesign | Some detail | Textual; activity list |

Continued

**Table 1** Continued

| Study name (year) | Publication type (name) | Hospital name | Simulation purpose | ED Detail | Patient Flow Description |
|---|---|---|---|---|---|
| Hay et al[31] (2006) | Conference proceedings (Winter Simulation Conference) | Unknown hospitals (n=4) | Understand behaviour | No detail | Flow chart; textual |
| Komashie and Mousavi[32] (2005) | Conference proceedings (Winter Simulation Conference) | Unknown hospital in London | Service redesign | More detail | Flow chart |
| Lane et al[33] (2000) | Journal article (Journal of Operational Research Society) | Unknown teaching hospital in London | Service redesign; forecasting | More detail | Flow chart; textual |
| Lattimer et al[14] (2004) | Journal article (Emergency Medicine Journal) | Nottingham City Hospital and QMC in Nottingham | Service redesign; forecasting | Some detail | Flow chart |
| Maull et al[34] (2009) | Journal article (The Service Industries Journal) | Unknown hospital in South West of England | Service redesign; forecasting | No detail | Flow chart |
| Meng and Spedding[35] (2008) | Conference proceedings (Winter Simulation Conference) | Unknown hospital | Service redesign | More detail | Flow chart; textual |
| Mould et al[36] (2013) | Journal article (Health Systems) | Unknown hospital in Fife, Scotland | Service redesign | No detail | Flow chart |

ED, emergency department; QMC, Queen's Medical Centre.

et al,[36] stakeholders were involved in deciding the study questions, specifying the model structure and implementing the model outputs. More than 80% (n=17) of the studies carried out some form of validation, mainly face and/or data-led validation. In face validation, project team members, potential users and other stakeholders subjectively compare model and real-life behaviours to judge whether the model and its results are reasonable at 'face value.'[41] Data-led validation involves the comparing of model output with 'real world' data and may also include a sensitivity analysis to determine the effect of varying the model's inputs on its output performance.[42]

Table 4 describes simulation results, summarising conclusions in terms of whether the changes considered (eg, increase in staffing numbers) were supported by the simulation, whether the changes supported were implemented in practice (eg, staffing increased), and barriers to conducting the simulation (eg, data issues) and implementing the changes supported (eg, poor clinician buy-in and credibility). Two-thirds of the studies (n=14; 67%) provided some discussion on the usefulness of simulation for analysis of changes to the delivery of emergency care (table 4): six studies supported the proposed changes, one study opposed the proposed changes and seven studies recommended differential changes. Only a small number of studies[12 34 36] (n=3; 14%) reported that the proposed changes supported by the simulations were implemented. For example, Maull et al[34] estimated the

impact of introducing a 'see and treat' strategy to reduce patient waiting times in the ED. After implementation, the observed reduction in breaches of the 4-hour waiting time target closely mirrored the simulation model predictions.

We identified a broad range of challenges, including oversimplified assumptions[22 25 33 35] and model structure,[14 25] system complexity,[11 14 30 31 34] poor data quality,[12 25 29 34 36] high expectations,[24] short-timescale,[33] poor stakeholder engagement,[12] limited specialist analytical skills,[36] model runtime,[11 24] generalisability[14 28] and impact of simulation[36]; six studies[10 21 23 26 27 32] (29%) did not describe any emergent issues.

## DISCUSSION

This review has shown that computer simulation has been used to analyse ED patient flow and resource capacity planning to the delivery of emergency care. The most common types of computer simulation used were DES (n=19; 90%) and SD (n=2; 10%). All but one study[25] provided either explicit or implicit justification for the choice of modelling method used. However, the use of computer simulation of patient flow within EDs in the UK does not appear to have increased in recent years as may have been expected. This could be a reflection of the relatively limited availability of funding for research in this area compared with funding for health technology

**Table 2**  Summary of simulation methods

| Study name | Simulation type | Rationale for simulation type | Model type | Simulation duration | Warm-up period | Simulation run | Simulation software |
|---|---|---|---|---|---|---|---|
| Anagnostou et al[21] | DES* | Yes | Specific | Not reported | Not reported | Not reported | Repast Simphony |
| Au-Yeung et al[22] | DES† | Yes | Specific | Not reported | Not reported | 10 | Written in Java |
| Baboolal et al[23] | DES | Yes | Specific | Not reported | Not reported | Not reported | Simul8 |
| Bowers et al[24] | DES | Yes | Specific | Not reported | Not reported | Not reported | Not reported |
| Brailsford et al[10] | DES‡ | Yes | Specific | Not reported | Not reported | Not reported | Simul8 |
| Coats and Michalis[25] | DES | No | Specific | Not reported | Not reported | Not reported | Simul8 |
| Codrington-Virtue et al[26] | DES | Yes | Specific | 52 weeks | Not reported | Not reported | Simul8 |
| Codrington-Virtue et al[27] | DES | Yes | Specific | 24 hours | 24 hours | 50 | Simul8 |
| Coughlan et al[28] | DES | Yes | Specific | 3 weeks | Not reported | Not reported | Simul8 |
| Davies[29] | DES | Yes | Specific | Not reported | Not reported | Not reported | Simul8 |
| Eatock et al[11] | DES | Yes | Specific | 3 weeks | Not reported | 20 | Simul8 |
| Fletcher et al[12] | DES | Yes | Generic | Not reported | Not reported | Not reported | Simul8 |
| Günal and Pidd[13] | DES§ | Yes | Generic | Not reported | Not reported | Not reported | Micro Saint Sharp |
| Günal and Pidd[30] | DES¶ | Yes | Generic | 52 weeks | 0 | 50 | Micro Saint Sharp |
| Hay et al[31] | DES | Yes | Specific | Not reported | Not reported | Not reported | Arena |
| Komashie and Mousavi[32] | DES | Yes | Specific | Not reported | Not reported | Not reported | Arena |
| Lane et al[33] | SD** | Yes | Specific | 24 hours | Not reported | 6 | iThink |
| Lattimer et al[14] | SD†† | Yes | Specific | 52 weeks | Not reported | Not reported | Stella |
| Maull et al[34] | DES | Yes | Specific | Not reported | Not reported | Not reported | Not reported |
| Meng and Spedding[35] | DES | Yes | Specific | Not reported | Not reported | Not reported | MedModel |
| Mould et al[36] | DES‡‡ | Yes | Specific | 3 months | 24 hours | Not reported | Simul8 |

*The authors used an agent-based simulation approach to model the ambulance service, but modelled the ED through a DES. These two individual models were then linked together to form a hybrid emergency services model.

†The authors used a Markovian queuing network, but computed the moments and densities of patient treatment time through a DES.

‡The authors used an SD model as part of a bigger picture, but modelled the ED through a DES.

§The authors used their ED model elsewhere[37] to form a whole hospital DES model consisting of two other departments: inpatient and outpatient clinics.

¶The authors used their ED model elsewhere[38] to form a whole hospital DES model consisting of three other components: inpatient bed management, waiting list management and outpatient clinics.

**The authors used their ED model elsewhere[39] to explore the issues that arise when involving healthcare professionals in the process of model building.

††The authors constructed the ED as a separate submodel which was not detailed in the paper. However, we believe this ED submodel[14] is identical to the ED model reported in another included study.[10]

‡‡The authors used their ED model elsewhere[40] to illustrate the role of care pathways to the redesign of healthcare systems.

DES, discrete event simulation; ED, emergency department; SD, system dyamics.

**Table 3** Detail of simulation inputs and outputs

| Study name | Source of arrival rates | Source of activity duration | Source of activity progression | Source of resources use | Changes considered | Outcomes considered | Validation | Stakeholder input |
|---|---|---|---|---|---|---|---|---|
| Anagnostou et al[21] | Not described | Not described | Not described | Not described | None | Waiting times | None | None |
| Au-Yeung et al[22] | Primary | Primary; expert opinion | Primary; expert opinion | Primary | ED patient flow | Waiting times | Data led | Model specification |
| Baboolal et al[23] | Primary | Primary; expert opinion | Primary | Primary | Resources | Waiting times*; resources | Face; Dark world model | Model specification |
| Bowers et al[24] | Primary | Primary | Primary | Not described | None | Resources* | Data led; Face | Model specification |
| Brailsford et al[10] | Primary | Secondary | Not described | Primary | ED patient flow; arrival rates | Waiting times | Data led; Face | Study question; model specification |
| Coats and Michalis[25] | Primary | Primary | Not described | Not described | Shift patterns | Waiting times* | Data led | None |
| Codrington-Virtue et al[26] | Primary | Not described | Not described | Not described | None | Resources | None | None |
| Codrington-Virtue et al[27] | Primary | Primary; expert opinion | Primary | Primary; expert opinion | None | Waiting times | Data led | None |
| Coughlan et al[28] | Primary | Not described | Primary | Primary | Resources | Waiting times*; resources | Data led | None |
| Davies[29] | Primary | Primary | Primary | Primary | ED patient flow | Waiting times* | None | None |
| Eatock et al[11] | Primary | Primary; expert opinion | Primary; expert opinion | Primary | None | Waiting times* | Data led | None |
| Fletcher et al[12] | Primary | Secondary; expert opinion | Primary | Primary | ED patient flow; resources; demand | Waiting times*; resources | Data led; Face | Study question; model specification |
| Günal and Pidd[13] | Primary | Not described | Not described | Not described | None | Waiting times* | None | None |
| Günal and Pidd[30] | Primary | Primary | Primary | Not described | ED patient flow; resources | Waiting times* | Data led | Model specification |
| Hay et al[31] | Primary | Not described | Not described | Not described | None | Waiting times* | Data led | None |
| Komashie and Mousavi[32] | Primary | Primary; expert opinion | Primary; expert opinion | Primary; expert opinion | ED structure; resources | Waiting times; resources | Data led; Face | Study question; model specification |
| Lane et al[33] | Primary | Primary; expert opinion | Expert opinion | Primary | Resources; demand | Waiting times; resources; elective cancellations | Data led; Face | Study question; model specification |

Continued

**Table 3** Continued

| Study name | Source of arrival rates | Source of activity duration | Source of activity progression | Source of resources use | Changes considered | Outcomes considered | Validation | Stakeholder input |
|---|---|---|---|---|---|---|---|---|
| Lattimer et al[14] | Primary | Primary | Primary | Primary | ED structure; admission rates | Bed occupancy | Data led Face | Study question; model specification |
| Maull et al[34] | Primary | Primary; expert opinion | Primary | Not described | ED patient flow | Waiting times* | Data led | Result implementation |
| Meng and Spedding[35] | Primary | Primary | Not described | Not described | ED structure; resources | Waiting times; resources | Data led Face | Study question; model specification |
| Mould et al[36] | Primary | Primary | Primary | Not described | Resources | Waiting times*; resources | Data led | Study question; model specification; result implementation |

*These studies used 4-hour target breach as part of their waiting time considerations.
ED, emergency department.

assessment. There is also a limited number of research groups with the analytical skills required to develop technically complex simulation models for the analysis of service redesign.

Identified studies varied in the style and quality of reporting; but assumptions used in the analyses were not always transparently reported. The opaque reporting of key assumptions prevents decision makers from appraising the quality of evidence from simulation experiments. Although there is a set of guidelines for researchers of DES to follow when building models,[43] this has not been widely adopted yet. Most of the studies (n=19; 90%) considered patient waiting or throughput times as the main outcome measure. This is perhaps unsurprising since waiting time has been shown to be a key determinant of patient satisfaction and has been strongly prioritised through the 4-hour targets.[2] Some studies[13 21 26 31] did not provide enough information on how input parameters were selected and synthesised. A handful of studies used expert opinion to populate some of the model inputs, but none explicitly justified the reason for using expert opinion. It is important to have transparent criteria for using expert opinion since it can overestimate or underestimate the model inputs. There are several methods for eliciting expert opinion as discussed by Grigore et al.[44]

Most models were intended to capture specific aspects of the emergency care process, but some authors have argued that understanding of patient flow requires study of the entire care process.[45] Conversely, others argue that it is sufficient to focus on the specific needs of the care process rather than modelling a large and complicated care process.[27] Most of the studies (n=18; 86%) used specific hospital models of ED patient flow. Interestingly, there seemed to be no standard hospital model of patient flow of emergency care process. One generic model was developed by the Department of Health in 2007 for use across all EDs.[12] This generic approach allows hospitals to benefit from simulation methodology with minimal costs and technical expertise, but there are challenges of using a generic national model for specific local use due to the local context of each NHS hospital including differences in physical space, the demographics of local patient populations, and so on.

Just over half of the studies in our review reported some involvement of stakeholders in the simulation study. Involving stakeholders is important since it helps to understand the problem better,[8 46] assess the simulation outputs fully[47] and translate simulation outputs into policy.[15] Very few studies reported clear summaries of whether the changes considered were supported by the simulation and of whether the changes supported were implemented. Some studies drew attention to a number of challenges particularly associated with simulation conduct and implementation. Brailsford[48] provided a helpful discussion on how to overcome the barriers such as methodological suitability, data crisis and stakeholder issues.

**Table 4** Summary of simulation results

| Study name | Conclusions | Conclusions detail | Reported the changes implemented? | Result implementation | Barriers |
|---|---|---|---|---|---|
| Anagnostou et al[21] | None | None | No | NA | None |
| Au-Yeung et al[22] | Supported the changes considered | Prioritisation of treatment for patient with minor problems over major problems could lead to improved outcome | No | NA | Simplified assumptions |
| Baboolal et al[23] | Supported the changes considered | A change in staffing levels could lead to substantial cost savings and reduce the 4-hour breaches | No | NA | None |
| Bowers et al[24] | None | None | No | NA | Model runtime; high expectancy |
| Brailsford et al[10] | Opposed the changes considered | Streaming of patients by triage category was not an efficient use of clinical resources | No | NA | None |
| Coats and Michalis[25] | Supported the changes considered | Shift pattern that best matches patient arrivals would give shorter waiting times | No | NA | Simplified model structure and assumptions; poor data quality |
| Codrington-Virtue et al[26] | None | None | No | NA | None |
| Codrington-Virtue et al[27] | None | None | No | NA | None |
| Coughlan et al[28] | Proposed differential changes | Adding an emergency nurse practitioner would not reduce the waiting times. Resource reallocation would improve throughput times | No | NA | Generalisability |
| Davies[29] | Supported the changes considered | The separation of see and treat would be beneficial | No | NA | Poor data quality |
| Eatock et al[11] | None | None | No | NA | System complexity; model runtime |
| Fletcher et al[12] | Proposed differential changes | Deflecting demand away from A&E would lead to improvement around waiting for beds, specialists and assessment processes | Yes | Unknown as other interventions were introduced in parallel | Poor data quality; poor stakeholder engagement |
| Günal and Pidd[13] | None | None | No | NA | Explaining the causes of change in performance |
| Günal and Pidd[30] | Proposed differential changes | More senior doctors, less X-ray requisitions and more cubicles would reduce waiting times | No | NA | Modelling multitasking behaviour of staff |
| Hay et al[31] | None | None | No | NA | System complexity |

Continued

**Table 4** Continued

| Study name | Conclusions | Conclusions detail | Reported the changes implemented? | Result implementation | Barriers |
|---|---|---|---|---|---|
| Komashie and Mousavi[32] | Proposed differential changes | Adding a nurse or doctor to minors would reduce the waiting times by 28%. Increasing the cubicles/beds would make smaller change | No | NA | None |
| Lane et al[33] | Proposed differential changes | Changing bed numbers led to no noticeable change in waiting times but a substantial difference to elective cancellations | No | NA | Short timescale; simplified assumptions |
| Lattimer et al[14] | Proposed differential changes | System would not be able to cope with increasing demand from scenario 1*, but scenarios 2†, 3‡ and 4§ could improve this | No | NA | Simplified model structure; system complexity; generalisability |
| Maull et al[34] | Supported the changes considered | See and treat reduced the 4-hour breaches from 13.2% to 3.4% | Yes | Marked reduction in no. of breaches from 13.2% to 1.4%. No. of patients waiting less than 1 hour increased from 12% to 23%. No. of patients with major problems waiting between 3 and 4 hours increased | Poor data availability and quality; system complexity |
| Meng and Spedding[35] | Proposed differential changes | Reduced times to see a consultant would reduce the waiting times. Access to 24-hour X-ray would reduce the waiting times too | No | NA | Simplified assumptions |
| Mould et al[36] | Supported the changes considered | A new staff roster would reduce the waiting times | Yes | Mean time for minor problems dropped from 100 to 94 min, for major problems it dropped from 200 to 195 min. Mean time for minor problems fell by 16 min after adjusting other factors | Poor data quality; limited analytical skills; impact of simulation |

*Five-year model run assuming 4% year-on-year growth in emergency admissions and 3% year-on-year growth in general practitioner (GP) referral for planned admissions.
†Impact of increase in demand for front door services.
‡Reducing emergency admissions of patients with respiratory or coronary problems, ill-defined conditions and over 65 years.
§Effects of earlier discharge of patients admitted as emergencies and subsequently discharged to nursing or residential homes.
A&E, accident & emergency department; NA, not applicable.

Only three studies[12 34 36] (14%) reported on the implementation of the changes supported by the simulation outputs. This may show that the impact of computer simulation modelling within the field of UK's emergency care has been limited, though we do not know if any changes were implemented at a later date. We also do not know if any changes implemented led to any improvements in the process or outcomes of ED care. The systematic use of simulation modelling is not yet part of healthcare, whereas its use in other sectors like in manufacturing or airline industry is an integral part of the actual decision-making process.[49] Why is simulation yet to make the same impact in healthcare as in other industries? Lack of stakeholders' engagement has been argued as one of the main reasons for this.[8 46 48 49] To this end, Harper and Pitt[46] discussed the basic components of successful implementation of simulation methods in healthcare. Absence of lucid guidelines about how to use simulation methods effectively in healthcare has been argued as another reason.[50] However, more recently in 2012, the ISPOR-SMDM Modeling Good Research Practices Task Force-4 laid out a set of guidelines about how to use DES method effectively in healthcare.[43] In line with a few others,[8 46 48 49] we also argue that if simulation is to make sustained impact in healthcare, the clinicians and decision makers must cooperate across physical and organisational boundaries and come to understand how seemingly small changes in design of processes can improve patient care.

We systematically searched eight bibliographic databases to identify the included studies; however, our study has some limitations. First, we focused on the use of computer simulation methods in the context of patient flow within EDs under the jurisdiction of UK NHS only. Improving emergency care is a research priority for UK NHS.[1] In this review, we examined the current literature that analysed ED patient flow within the context of UK, and discussed how simulation can be better used as a tool to address this problem. It would be interesting to compare the identified methods with other jurisdictions across Europe, in the USA and Australasia, but this was beyond the scope of this study. Besides, comparing studies from different jurisdictions and reaching consensus would be challenging since healthcare delivery is different in the UK. Nevertheless, computer simulation has been used to analyse and design ED overcrowding in other countries. In particular, DES models have been used to identify optimal ED flow patterns,[51] forecast ED overcrowding[52 53] and evaluate staffing levels and changes in ED bed capacity.[54] Fletcher et al[12] cited a number of other international ED models which have different designs to English ED.

Second, we were not aware of any formal assessment checklist to estimate quality scores of the identified studies. The set of guidelines reported by the ISPOR-SMDM is not a quality assessment checklist for reviewers.[43] It is rather a set of recommended best practices for modelling teams to consider and embrace when building DES models. Furthermore, there is a good rationale for a component-based approach, instead of using a quality score. For example, in the field of randomised controlled trials (RCTs), there is evidence that the use of quality scores and scales, especially of those with a numerical summary, is problematic and meaningless.[55] The current best practice for assessment of validity of RCTs requires assessing individual components of trial design, conduct and analysis (eg, Cochrane risk of bias tool). We adopted a similar approach, whereby assessing the key methodological components of all included studies.

Third, we neither verified whether any of the hospitals implemented the findings found from simulation experiments, nor do we know if any changes implemented led to any improvements. Typically, there is little opportunity to assess the impact of the simulation since publication emerges before the work is fully implemented in many healthcare studies.[56] Finally, we did not include Google Scholar in the database search list since it has a number of issues with its indexing and citation algorithm, although it is known to provide increased access to non peer-reviewed publications.[57] Anecdotal evidence suggests that NHS hospitals have used simulation modelling (and other methods) to improve patient flow through the ED. However, our review will not capture all of this work as it is not all reported in peer-reviewed academic publications. We used a key review paper[20] to select a wide range of databases covering the comprehensive sources of literature in computer science, operations management and healthcare fields.

## CONCLUSIONS

We found that computer simulation can provide a means to pretest the likely impact of changes to the delivery of emergency care before implementation in a safe and efficient manner. In particular, it is used to identify the key relationships and bottlenecks in the process of ED care, test 'what-if' scenarios for service redesign, determine levels of uncertainty, provide visualisations and forecast future performance. However, the evidence base is small and poorly developed, with many methodological and practical issues, including lack of awareness regarding system complexity, lack of good quality data, lack of persistent engagement of stakeholders in the modelling process, lack of in-house analytical skills and lack of an implementation plan. Furthermore, the level of detail of reporting of the computer simulation methods differed in the style and quality of reporting; and in some instances, key aspects of the assumptions underpinning the analyses were not always reported explicitly and transparently.

This review is a useful source providing direction on why simulation needs to be better used as a tool for analysis of ED patient flow. Future studies should justify the choice of simulation modelling method explicitly, avoid making selective use of the available data, engage stakeholders in the modelling process and keep them on board continually, be transparent in the reporting of simulation inputs and outputs, and report on the implementation of changes supported by the findings of

simulation experiments. We recommend the adoption of reporting guidelines[43] by academic journals and conference proceedings, and more persistent exploitation of innovative models of engagement and knowledge mobilisation between academics and healthcare professionals such as the Researchers in Residence.[58] Further research is necessary to assess the quality of computer simulation models of ED patient flow across different countries and to establish the extent to which the simulation outputs have been translated into policy.

**Contributors** SM and JB independently screened and appraised the relevant studies, and extracted the data from the included studies. CV, SM, JS, KN, JB and AR designed the search strategy. AR, an information specialist, conducted the electronic search. SM drafted the manuscript in conjunction with CV. JD, WH and CV revised the draft critically for intellectual content, while the other authors commented on the draft. All authors read and approved the submitted manuscript.

**Funding** All the authors apart from CV are supported by the National Institute for Health Research (NIHR) Collaboration for Leadership in Applied Health Research and Care West (CLAHRC West) at University Hospitals Bristol National Health Service (NHS) Foundation Trust. The views expressed are those of the authors and not necessarily those of the NHS, the NIHR or the Department of Health.

**Competing interests** None declared.

**Provenance and peer review** Not commissioned; externally peer reviewed.

**Data sharing statement** No additional data available.

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
