## [Reviewer comments · BMJ Open]

ARTICLE DETAILS

TITLE (PROVISIONAL)	A systematic review of the use of computer simulation modelling of patient flow within UK emergency departments
AUTHORS	Mohiuddin, Syed; Busby, John; Savovic, Jelena; Richards, Alison; Northstone, Kate; Hollingworth, William; Donovan, Jenny; Vasilakis, C

VERSION 1 - REVIEW

REVIEWER	T. Eugene Day, D.Sc. Children's Hospital of Philadelphia, United States.
REVIEW RETURNED	13-Dec-2016

GENERAL COMMENTS	A new systematic review of simulation studies in the Emergency Department is most welcome. This paper does a good job of describing the purpose and utility of DES for modeling the ED, and mentions Systems Dynamics as well. Overall, the authors appear to do good work in identifying and categorizing the relevant literature. However, some limitations to the paper exist that, if addressed, would improve its impact. 1. Why was Google Scholar not included in the search database list? While there are certainly some issues with GS's indexing and citation algorithms, it tends to provide comprehensive search results. I would recommend the authors update their search list using this engine.2. The authors report that they used (page 6, line 31) "a wide range of search terms", and that they "developed a strategy" for each search engine. The specific search terms would be a useful appendix, so that we could evaluate its completeness. And more detail about the MEDLINE search strategy which is included would be useful to interpret the jargon.3. A quibble: random variables can certainly be used in systems dynamics models, but in a wholly different way from how they are used to represent the diversity of individual patients and processes in a DES model. The blanket statement that RVs "cannot be used" in SD is overly broad, and misleading. The author's intent is clear (and reasonably accurate), but the language is not quite right.4. The limitations discussed need to be expanded and included in the discussion more concretely. If the authors choose not to include Google Scholar, it should be mentioned here. While I completely understand the restriction of scope to the NHS, important points might be noted that restricting the conclusions to DES's performed in the NHS (and the ED) context restricts the strengthening of those
--

	conclusions rather markedly: DES has been used to support implementation of positive change robustly in other places, and other clinical environments. 5. The authors note that they are unaware of standards for the development of DES in a healthcare environment. I would encourage the authors to consider the series of papers from the Society of Medical Decision Making's modeling good research practices task force in 2012, which lay out a set of recommended best practices which many researchers and practitioners of DES consider and embrace when building models. The reference the authors use to support their claim is from 2009. Overall, this is a clear and high quality manuscript which will benefit from a broader perspective in the discussion (if not the methods), and more recent supporting references. I appreciate the opportunity to review it.
--	---

REVIEWER	Tillal Eldabi Brunel University UK
REVIEW RETURNED	30-Dec-2016

GENERAL COMMENTS	Accident and Emergency departments or Emergency Departments (ED) as denoted in this article, are the most common target of simulation studies. This may be due to the nature of EDs, being closed systems with well understood processes and quantifiable outputs. The subject of ED simulation requires more studies and reviews. Given the vast number of literature on modelling and simulation of EDs, there is little evidence of outcomes improvement. Therefore, I find the principle of conducting such a review very topical. On the other hand, the paper is disappointing in many aspects. The article only found 21 relevant studies of EDs in the UK. This is a small number considering the overall number of known studies. This is not a major issue but it would raise concerns about the coverage of this review. On the other hand, the paper provides a significant introduction about discrete event simulation (DES) and system dynamics (SD) whilst ignoring other approaches such as agents based modelling, Monte Carlo simulation and hybrid simulation where we find that this review includes some agent/hybrid paper which were wrongly classified (see for example reference 19 Anagnostou et al 2013). The overall discussions about the benefits of simulation to EDs and the text about DES and SD are out of date. The "benefits" of simulation have been widely reviewed and well known. Currently, there is more need on how to implement simulation findings and how to engage stakeholders. These are current research questions that have been confirmed by this review as conclusions, where as a matter of fact these are well trodden issues. Although it maybe conceivable that focusing on ED modelling in the UK only will provide a better picture about supporting ED within the UK, it would be beneficial to open the review worldwide as EDs are mainly similar in many countries around the world. this would help gaining more lessons about ED simulation.
--

VERSION 1 – AUTHOR RESPONSE

Author's Response to Decision Letter for (bmjopen-2016-015007)

A systematic review of the use of computer simulation modelling of patient flow within UK emergency departments

School of Social & Community Medicine
University of Bristol
Whitefriars (9th Floor)
Bristol BS1 2NT

6 February 2017

Editor-in-Chief
BMJ Open

Dear Dr Groves

Manuscript ID bmjopen-2016-015007 entitled: "A systematic review of the use of computer simulation modelling of patient flow within UK emergency departments."

Many thanks for sending us the reviewers' comments on this manuscript. We have taken account of these comments, and amended the manuscript accordingly making the suggested changes. Below we have detailed how we responded to each of the reviewers' comments made. We have indicated where we have made some additional changes to the original manuscript submitted using yellow highlighted text.

Thank you very much for re-considering this manuscript in BMJ Open and we look forward to hearing your response.

Yours sincerely

Syed Mohiuddin, PhD
Research Fellow in Modelling and Simulation in Health Economic Evaluation
(On behalf of all authors)

EDITOR

Comment: Please revise/ remove the final point of the 'Strengths and Limitations' section on page 3. This is not a strength or a limitation of the study.

Response: We have removed the final point as advised.

Comment: Please further justify why you did not assess study quality, especially in light of reviewer 2's comment that he is surprised that you found only n=21 studies to include. We note that you say: "We did not assess the quality of the identified simulation models since we are not aware of any guidelines to carry out this type of quality assessment." Does that mean the studies themselves were not assessed? We note that on page 10 you say that there are guidelines for reporting inputs and outputs from DES modelling (reference 41).

Response: We have provided further justification (on Page 12) of why we did not assign a quality score to each paper. This however does not mean that we did not assess validity and methodological quality of included studies.

We assessed each study in terms of simulation method, key assumptions, input and output data, conclusions drawn, and implementation of results. However, we formally did not allocate each study with a quality assessment score since there is no such assessment checklist to follow.

There is a good rationale for a component-based approach, instead of using a quality score. For example, in the field of randomised controlled trials there is evidence that the use of quality scores and scales, especially of those with a numerical summary, is problematic and meaningless (Ref. 55) The current best practice for assessment of validity of RCTs requires assessing individual components of trial design, conduct and analysis (e.g. Cochrane risk of bias tool). We adopted a similar approach, whereby assessing the key methodological components of all included studies.

The ISPOR-SMDM guidelines (Ref. 43; previously 41) are not a quality assessment checklist for reviewers. It is rather a set of recommended best practices which many researchers consider when building DES models. We have rephrased the sentence on Page 11 (previously on Page 10). We have also provided further justification on Page 12.

REVIEWER 1

Comment 1: Why was Google Scholar not included in the search database list? While there are certainly some issues with GS's indexing and citation algorithms, it tends to provide comprehensive search results. I would recommend the authors update their search list using this engine.

Response: Google Scholar does not search a well-defined set of journals, while it provides increased access to non peer-reviewed publications (Ref. 57). The shortcomings of Google Scholar and its search interface have been well-documented in the literature and include lack of reliable advanced search functions, lack of controlled vocabulary and issues regarding scope of coverage.

We were interested in peer-reviewed academic publications only. Included databases are the comprehensive sources of literature in computer science, operations management, and healthcare fields. Our database search list was validated using a key paper (Ref. 20) known to us.

We have added a sentence on Page 6 to confirm that: "We also conducted backward and forward citation searches of all included studies using Google Scholar."

However, in line with the reviewer's recommendation in Point 4 below, we have provided reasons (on Page 13) of why we excluded Google Scholar from the primary database list.

Comment 2: The authors report that they used (page 6, line 31) "a wide range of search terms", and that they "developed a strategy" for each search engine. The specific search terms would be a useful appendix, so that we could evaluate its completeness. And more detail about the MEDLINE search strategy which is included would be useful to interpret the jargon.

Response: The MEDLINE search strategy has been provided in Appendix 1.

We have provided additional text on Page 6.

Comment 3: A quibble: random variables can certainly be used in systems dynamics models, but in a wholly different way from how they are used to represent the diversity of individual patients and processes in a DES model. The blanket statement that RVs "cannot be used" in SD is overly broad, and misleading. The author's intent is clear (and reasonably accurate), but the language is not quite right.

Response: We have edited this sentence on Page 5 in the following way:

"Unlike DES, system dynamics does not lend itself readily to including random variables and thus input parameters are given as simple rates in the majority of studies."

Comment 4: The limitations discussed need to be expanded and included in the discussion more concretely. If the authors choose not to include Google Scholar, it should be mentioned here. While I completely understand the restriction of scope to the NHS, important points might be noted that restricting the conclusions to DES's performed in the NHS (and the ED) context restricts the strengthening of those conclusions rather markedly: DES has been used to support implementation of positive change robustly in other places, and other clinical environments.

Response: We have provided additional text on Pages 12 and 13.

Comment 5: The authors note that they are unaware of standards for the development of DES in a healthcare environment. I would encourage the authors to consider the series of papers from the Society of Medical Decision Making's modeling good research practices task force in 2012, which lay out a set of recommended best practices which many researchers and practitioners of DES consider and embrace when building models. The reference the authors use to support their claim is from 2009.

Response: We have provided additional text on Page 12.

REVIEWER 2

Comment 1: The article only found 21 relevant studies of EDs in the UK. This is a small number considering the overall number of known studies. This is not a major issue but it would raise concerns about the coverage of this review.

Response: We conducted this review systematically. Two authors independently applied the selection, appraisal and data extraction processes. This review complies with the PRISMA checklist. Based on the inclusion and exclusion criteria laid out on Pages 6 and 7:

"A total of 21 studies^{10-14 21-36} were included in the final review. Four studies³⁷⁻⁴⁰ were excluded from the final review because the models used by these studies are identical to the models reported in other already included studies.^{13 30 33 36}"

We believe we have included all the studies which met our eligibility criteria.

Comment 2: On the other hand, the paper provides a significant introduction about discrete event simulation (DES) and system dynamics (SD) whilst ignoring other approaches such as agents based modelling, Monte Carlo simulation and hybrid simulation where we find that this review includes some agent/hybrid paper which were wrongly classified (see for example reference 19 Anagnostou et al 2013).

Response: There are several computer simulation modelling (CSM) approaches available. Our review shows that DES (an extension of Monte Carlo simulation) and SD methods were the only two that had been applied in analysis of NHS ED patient flow. Given that the goal of our review was to summarise CSM methods used to evaluate ED patient flow in the NHS, we opted not to discuss all possible CSM approaches. We have provided some additional text on Page 5 to better clarify these issues.

Anagnostou et al (2013) used two separate classes of models: (i) agent based simulation (ABS) for the 'ambulance service model' and (ii) discrete event simulation (DES) for the 'A&E model'. Later, these two individual models were linked together to form a hybrid emergency services model. Our review focused on the A&E care process only. We have provided a table footnote (see Table 2) to clarify the issue.

Comment 3: The overall discussions about the benefits of simulation to EDs and the text about DES and SD are out of date. The "benefits" of simulation have been widely reviewed and well known. Currently, there is more need on how to implement simulation findings and how to engage stakeholders. These are current research questions that have been confirmed by this

review as conclusions, where as a matter of fact these are well trodden issues.

Response: Little is known about the usefulness of computer simulation for analysis of ED patient flow. Therefore, we undertook this review to examine the different computer simulation methods and their contribution for analysis of patient flow within EDs in the UK. This review is a useful source providing direction on why simulation needs to be better used as a tool to address the problem.

We acknowledged that further research is necessary to establish the extent to which the simulation outputs have been translated into policy.

Comment 4: Although it maybe conceivable that focusing on ED modelling in the UK only will provide a better picture about supporting ED within the UK, it would be beneficial to open the review worldwide as EDs are mainly similar in many countries around the world. this would help gaining more lessons about ED simulation.

Response: On Page 12, we have now widened the discussion to further cite the international literature using computer simulation modelling in the ED.

Healthcare delivery is different in the UK (Ref. 12). Fletcher et al (Ref. 12) cited a list of non-English A&E models which have different designs to English A&E. Our objective was to provide a more in depth review of the literature most relevant to UK A&Es rather than a broader but shallower review of the international literature.

VERSION 2 – REVIEW

REVIEWER	Theodore Eugene Day Children's Hospital of Philadelphia
REVIEW RETURNED	20-Feb-2017

GENERAL COMMENTS	The authors have addressed my concerns.
---